



# Estimation of radar-based Area-Depth-Duration-Frequency curves with special focus on spatial sampling problems

Golbarg Goshtasbpour[1] and Uwe Haberlandt[1]

[1]Institute of Hydrology and Water Resources Management, Leibniz University Hannover

**Correspondence:** Uwe Haberlandt (haberlandt@iww.uni-hannover.de)

**Abstract.** Radar-based Area-Depth-Duration-Frequency (ADDF) curves offer the possibility of incorporating a space dimension into analysis of rainfall extremes. This solves some shortcomings of the traditional point-based Depth-Duration-Frequency (DDF) curves which characterize design rainfall. In this study, ADDF curves are calculated from a radar-based rainfall data set, a product of the conditional merging of corrected radar data and station data, covering a large area in north part of Germany. The
initial results show implausible behavior in the curves where the rainfall quantiles increase with increasing area. It is discussed in details in this paper that the implausible behavior persists due to the shortcoming of fixed-area sampling methods which is missing the most extreme annual maximum rainfall events within the area of interest. Three alternative sampling strategies are developed to address this issue. Among the introduced methods the Multiple-Location-Extreme-Sampling (MLES) and the Single-Location-Extreme-Sampling (SLES) methods successfully reduced the number of study locations with implausible
behavior by 67% and 43% respectively. The SLES method is recommended as the best method for calculating areal design rainfall directly from high resolution radar-based data sets. This method tackles the spatial sampling issue and it can result in Area-Reduction-Factor values compatible with station-based point design rainfall values.

## 1 Introduction

The frequency, intensity and spatial occurrence of extreme rainfall events are changing with global warming (Papalexiou and
Montanari, 2019). This has implications for catastrophic flooding, overloading storm water and sewage systems, soil erosion, landslides and damaging infrastructure. Most of the consequences of extreme rainfall are due to the sheer volume of water imposed on settlements and infrastructure, sometimes in a very short period of time. This shows the importance of studying rainfall and rainfall extremes as a spatial-temporal phenomenon. This task is challenging due to multi-dimensional nature of rainfall extremes and their high variability in space and time.

The Intensity-Duration-Frequency or Depth-Duration-Frequency (IDF/DDF) relationship, is a tool used to characterize frequency and magnitude of extreme rainfall events and are used for planning and design. The IDF relationship is typically presented as a set of curves and connects the intensity (or depth) of extremes with their duration and the frequency of occurrence or return period. IDF curves provide information solely for single points while for many applications areal extreme rainfall estimates are of interest. The conversion to areal precipitation is possible using Areal-Reduction-Factors (ARF), a corrective
factor defined as the ratio between the areal precipitation and the representative point precipitation of an area, for a specific





return period and duration, with the assumption that the larger the area becomes, the smaller the average areal precipitation depth should be. Many studies have used ARFs and IDF curves for different purposes like design storm characterization (Kim et al., 2019), design peak discharge (Bertini et al., 2020) , flood hazard quantification (Ghazavi et al., 2016) and development of early warning systems (Bezak et al., 2016). Another way of handling areal rainfall extremes, not as widely used as the ARF
method, is to apply extreme value analysis directly on areal rainfall, and calculate Intensity-Duration-Area-Frequency (IDAF) curves (Bennett et al., 2016; Panthou et al., 2014; Mélèse et al., 2019; Overeem et al., 2010) .

ARFs and IDAF curves can be calculated through different analytical and empirical methods, where the former focuses on the mathematically determined physical laws and the underlying characteristics of extreme precipitation, and the latter derives the IDAF and/or ARF relationship based on observation data. As more legitimate and reliable the analytical approaches
might appear, these methods involve assumptions which do not fully represent the truth of the rainfall extremes. Data-driven approaches on the other hand, although computationally more expensive, are based on the observations, which are the closest information to truth at hand (Svensson and Jones, 2010). Empirical methods themselves are divided into fixed-area and storm-centered categories. As the names suggest, the former calculates the precipitation for a fixed area and assumes a representative point rainfall (a specific point in the considered area or the average of the point extremes), whereas the latter, the point with
the maximum rainfall depth observed for a specific duration and area is not fixed and the areal precipitation is calculated for individual moving storms (Biondi et al., 2021).

Bennett et al. (2016) construct empirical IDAF curves from a rain gauge-based interpolation product and assess the characteristics of the extremes at 11 scattered study regions across Australia, covering a variety of climates. Panthou et al. (2014) calculate analytical IDAF curves, using a rain gauge-based data set and concluded that their ARFs approximate the empirical
ARFs of the region, except for events with short durations and large spatial scales. Bárdossy and Pegram (2018), however, demonstrate the shortcomings of the station-based extreme precipitation estimation through an extensive analysis. They show that IDF curves are only suitable for the single observation sites and cannot be expanded to larger areas around the observation station, since they do not reflect the spatial behavior of precipitation. They suggest that the maxima of areal precipitation may exceed single point extremes, thus ARFs might underestimate the design values. Results from Bennett et al. (2016) are an
example of a case where station-based areal extreme precipitation estimation leads to areal average extremes higher than single point extremes which according to Kim et al. (2019) is not representative of the reality of precipitation. Concluding from the statements of Bárdossy and Pegram (2018), Bennett et al. (2016) and Kim et al. (2019) there is a gap between what the extreme sampling methods capture and the true spatial characteristics of these events.

Radar rainfall estimates, which have become increasingly popular in hydrological research in the recent decades, offer
high spatial and temporal resolution and large coverage areas. Thus radar rainfall data improve capability to capture the spatial characteristics of rainfall relative to ground gauge networks. It is well known that radar data is error-prone, therefore, numerous methods have been developed to reduce biases associated with these errors for instance correction algorithms and merging methods (Krämer and Verworn, 2009; Sinclair and Pegram, 2005). Quality controlled radar-based precipitation data can be of great value as it enables the experts to develop methods which approach the extreme precipitation from a spatial perspective.
Lengfeld et al. (2020) showcase the importance and benefits of using radar data. They report that in year 2014 in UK 36%





of the hourly events and 50% of the daily events observed by radar were not captured by the rain gauges. The same analysis on records from Germany revealed between 2001 and 2018 only 17.3% of hourly and 81% of daily heavy precipitation was captured by the rain gauge network.

Overeem et al. (2010) offered the first radar-based study which calculates DDF curves for different durations and areas from The Netherlands using the GEV probability distribution of the annual maxima as a function of duration and area. Mélèse et al. (2019) constructed an IDAF analytical model for a region in the south of France using a radar reanalysis data set where they classified the shape of the resulting ARF curves to distinguish the spatial structures of the extremes in the region. Rosin et al. (2023) investigate the characteristics of the extremes in coastal, desert and mountainous regions covering the eastern Mediterranean. They apply the novel approach of Simplified Meta-statistical Extreme Value (Marra et al., 2019) on 12 years long radar precipitation data and calculate the corresponding IDAF curves for different temporal and spatial scales. Haruna et al. (2023), using a radar reanalysis data set, aim to take advantage of all available information at hand. Instead of working with block maxima, they model their IDAF curves by including all non-zero precipitation and fitting an extended generalized Pareto distribution. They take an empirical data-driven approach like Overeem et al. (2010) and study extreme precipitation in Switzerland, an area with a complex topography, multiple precipitation regimes and seasonal and regional variability. Zhao et al. (2023), explore the variability of the extreme rainfall events in space and time for current and future climate by building ARF-based IDAF curves using radar data. In their study the authors work with the peaks over threshold instead of annual maxima, calculate the ARFs and apply them to the future extreme events from the future projections.

In this study we offer an empirical method for calculating Area-Depth-Duration-Frequency (ADDF) curves based on the extreme value analysis of Koutsoyiannis et al. (1998) using a merged radar data set covering a part of north-west of Germany. More importantly we investigate the spatial order relation problems, appearing as crossings in ADDF curves, which lead to missing information in areal rainfall extreme value analysis and underestimation of design storms. For this purpose we define novel measures to quantify the issue that appears in the spatial order relation of the curves and look at the relation between the crossings and topography and seasons. We investigate alternative for sampling approaches which can lead to practical solutions to this problem and compare their results with standard design storms for the region. To our knowledge, there are no studies investigating the spatial order problem in detail and offering new sampling methods.

In section 2 the study area and different data used are introduced and the quality of merged radar data set is proved by a cross validation against ground stations. Section 3 describes our different sampling methods of areal extremes and explains the mathematical framework of the extreme value analysis. Section 4 presents the results which are discussed in detail in Section 5. Finally the findings and conclusions are expressed in section 6.

## 2 Study Area and Data

### 2.1 Study Area

The study area is located in Northern Germany, covered by the range of Hanover radar, in Lower Saxony. The region has been observed for the time period of 2000 to 2019, using data from radar, daily and 5-min rain gauge data sets provided by



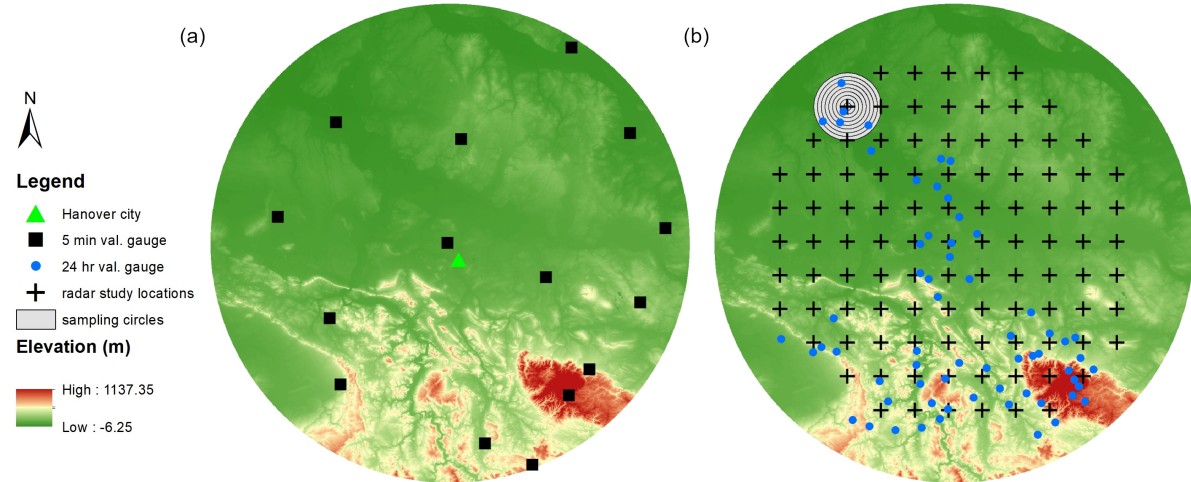

**Figure 1.** Study area, validation stations, investigated locations and covered areas

the German Weather Service (DWD). The radar beam is 128 km long and covers an area of roughly 50,000 km$^2$ (Fig. 1).

The northern segment, a part of the North German Plain, is predominantly flat, while the southern part is home to the Harz Mountains in the southeast, peaking at 1137 m.a.s.l. Annual precipitation in the study area vary from 500 mmyr$^{-1}$ to 1700 mmyr$^{-1}$, with the highest recorded amounts occurring in the Harz Mountains.

## 2.2    Data

### 2.2.1    Rain gauge data

The station data sets used in this study include: a) a 5-min station network used for adjusting the radar data by merging the two data sources and also to validate the final merging radar data and b) a daily station data set with a higher spatial density, used as reference to compare the ADDF curves in the results section. From the available 5-min stations only 15 station were picked out as the validation station which had the highest rate of data availability in the study time period (Fig.1, A). From the daily stations, the ones are chosen which have high data availability within the study period and build a dense network of at least 4

stations around each of the radar study locations. Due to the scattered availability of the daily stations only 19 radar locations and 63 daily station are chosen for a comparison of the spatial analysis of the radar data with the daily stations (Fig.1, B).

### 2.2.2    Radar data

The Hanover radar device is a C-band instrument operating at a 5 min temporal resolution with a 1° azimuth resolution and a spatial resolution of 1 km along the beam. This study utilizes the dx product from DWD with a scanning radius of 128

km and 1 km resolution. Each radar image of the dx product consists of 46,080 cells (128×360) and the data is offered in reflectivity values which were not corrected in any manner. The data is then preprocessed and corrected for clutter and





attenuation according to the Berndt et al. (2013) method using the tools provided in the *wradlib* Python library (Heistermann et al., 2013).

The preprocessing procedure begins with generating a static clutter map for each year. This involves scanning every time step in the year, identifying pixels with extremely high or low values, and marking them as clutter, resulting in a map with zeros and ones indicating clutter pixels. Following this, a comprehensive scan is conducted over all time steps to detect dynamic clutter. The static and dynamic clutter maps are then combined, and the values at cluttered pixels are replaced with interpolated values from neighboring pixels using an inverse distance method. Subsequently beam attenuation correction and constrained attenuation correction are applied to each time step. Finally, reflectivity values are transformed into rainfall depth using the Marshall-Palmer Z–R relationship

$$Z = aR^b \tag{1}$$

where Z represents reflectivity in $\mathrm{mm^6 m^{-3}}$, R is the corresponding rain intensity in $\mathrm{mm h^{-1}}$, and the parameters a and b, set to 256 and 1.42 respectively per DWD's standard guidelines, are used. The polar grid is then converted into a 1 km × 1 km Cartesian grid within the Gauss-Kruger-3 coordinate system. After the initial correction, a copula-based conditional merging is applied to combine the radar and rain gauge observation (El Hachem, 2023).

### 2.2.3 Merged Data Validation

The final merged product is validated by comparing performance measures of the corrected radar product and the merged products. For this purpose, merging is done in a leave-one-out cross-validation mode: to calculated the merged value at a specific validation station, the data from that station is left out and the merged time series are calculated for that location. Then the DDF curves are calculated from the corrected radar data, merged data and the station data at the validation station using the Extreme Value Analysis (EVA) method explained in the Methodology section.

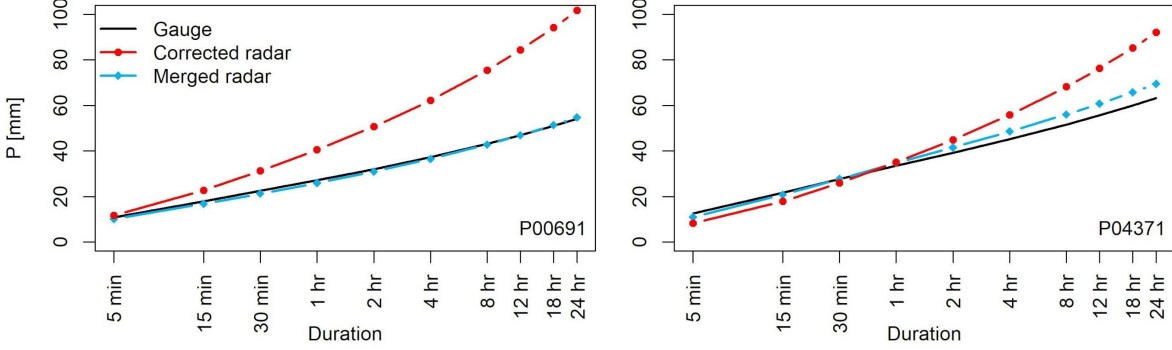

**Figure 2.** Example IDF curves of two validation stations P00691 and P04371, curves from three sources are displayed: Ground rain gauges (black), corrected radar data (red) and the product of conditional merging (blue)





The performance measures used for validation are the percent-bias (pBias)

$$pBias = \sum_{i=1}^{n_{gauge}} \frac{P_{radar,i} - P_{gauge,i}}{P_{gauge,i}} \tag{2}$$

and the normalized root-mean-squared-error (nRmse)

$$nRmse = \frac{\sqrt{\sum_{i=1}^{n_{gauge}} (P_{radar,i} - P_{gauge,i})^2}}{\sum_{i=1}^{n_{gauge}} P_{gauge,i}} \tag{3}$$

which are calculated between the station DDF curves and the curves from the radar data and merged data at the location of the validation stations.

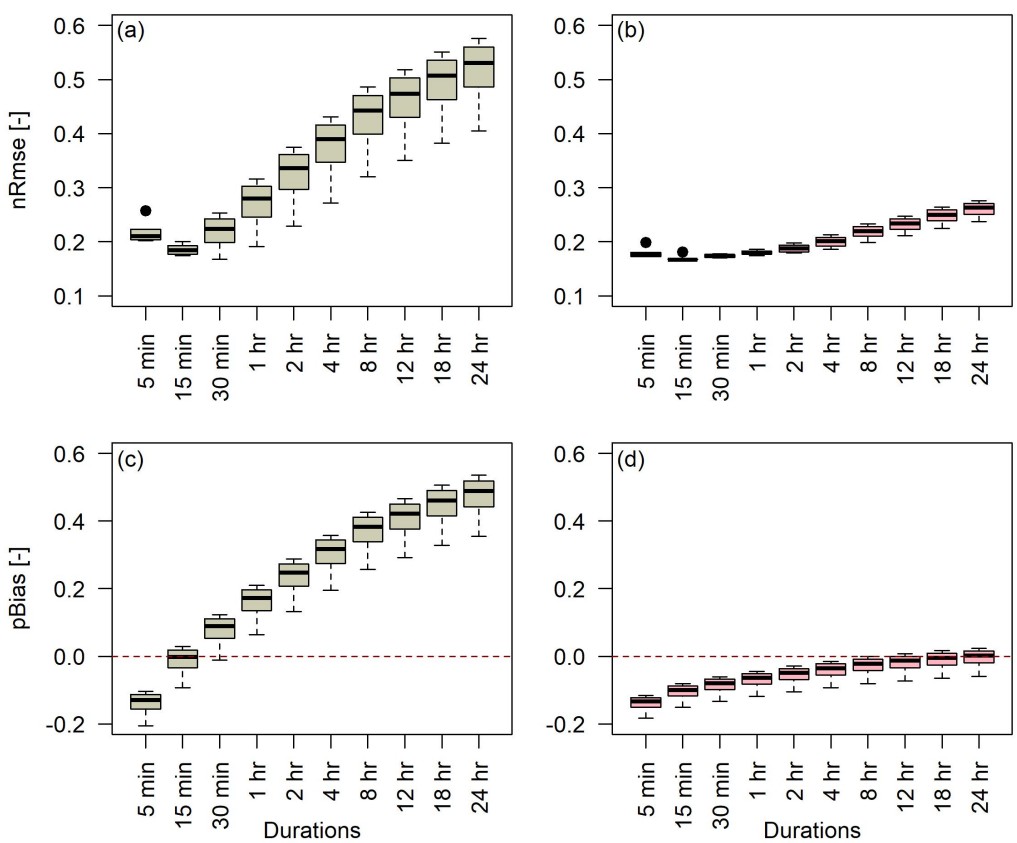

**Figure 3.** Error measures of the point percentiles of the corrected radar product (A and C) and the conditional merging product (B and D). The box plots show the statistics of error over validation stations for different return periods (2, 3, 5, 10, 20 and 33 yr)

Figure 2 depicts two examples of the DDF curves, where a significant improvement is observed after merging. The performance measures are subsequently calculated for each duration and return period over the 15 validation stations and the results





are depicted in Fig. 3, where the panels A and B show the nRmse and C and D show the pBias, left column representing the corrected radar results and right column the merged data. The box-plots show the range of the error over different return periods. These figures show a significant reduction of both error measures after merging. With an average error of nRmse of 0.2 and pBias of -0.06 it is shown that the merged product performs well in estimating rainfall extremes at point level, thus it can be used for this study.

## 3  Methodology

The ADDF curves are calculated from the merged radar data. We define 100 equidistant study locations in space (Fig. 1). For each location:

1. Areas around the study location are considered in circular shapes with radii of: r = 1, 2, 4, 6, 8, 10, 12, 14, 16 and 18 km (area of a = 1, 12, 50, 113, 210, 314, 452, 615, 804, 1017 $\text{km}^2$) where the r = 1 km is equivalent to one pixel and is considered as a point. Areal rainfall $P_a$ is calculated for each area as the arithmetic mean over all pixels k = 1, 2, 3, ..., $n_a$:

$$P_a = \frac{\sum_{k=1}^{n_a} P_k}{n_a}. \tag{4}$$

2. The areal rainfall series are aggregated in time using a moving window for durations d = 5, 15, 30, 60, 120, 240, 480, 720, 1080 and 1440 min (Eq. (5)) and the annual maxima series for m years is extracted for each area and duration by taking the maximum value of the temporally aggregated series for each year, defined as $AM_a(d)$ (Eq. (6)):

$$P_{a,t}(d) = \sum_{j=t}^{t+d} P_{a,j} \tag{5}$$

$$am_{yr}(a,d) = \max_{yr} P_{a,t}(d) \tag{6}$$

$$AM_a(d) = am_1(a,d), am_2(a,d), ..., am_m(a,d) \tag{7}$$

3. Extreme value analysis according to Koutsoyiannis et al. (1998) is applied and the quantiles are calculated for each area, duration and return period T = 2, 3, 5, 10, 20 and 33 yr; $P_T(a,d)$

4. ADDF curves are plotted in $\text{P}_\text{T}[\text{mm}]$ vs. $\text{Duration}[\text{min}]$ in a half-logarithmic plot, as a group of curves from different area sizes.

This is the overall procedure used here for calculating the ADDF curves, however as new sampling alternatives are investigated and discussed in subsections 3.2.2. to 3.2.4. These steps are slightly modified for each sampling technique. In the following subsections the EVA method is explained, the sampling methods are introduced and finally new measures are introduced to facilitate the quantification of the observed spatial sampling problem in the results.



## 3.1 Extreme Value Analysis

The extreme value analysis method developed by Koutsoyiannis et al. (1998) was employed here. In this approach the intensity of the annual maxima is considered to be a function of the duration as follows:

$$i_{a,d} = AM_a(d)/d \tag{8}$$

with

$$i = i_d.b_d \tag{9}$$

and

$$b_d = (d + \theta)^\eta \tag{10}$$

where $i$ is the *generalized intensity* of the annual maxima in $\mathrm{mmhr}^{-1}$, $i_d$ the observed rainfall intensity in $\mathrm{mmhr}^{-1}$ with the duration d. $\theta$ and $\eta$ are the Koutsoyiannis transformation parameters where $\theta$ has values larger than 0 and $\eta$ takes any value between 0 and 1. The two Koutsoyiannis parameters ($\theta$ and $\eta$) are calculated for each series of annual maxima through an op-

timization process where the Kruskal-Wallis statistic comparing the distribution for different duration is minimized. For more details the reader is referred to the original paper Koutsoyiannis et al. (1998). With $\theta$ and $\eta$ determined, the annual maximum intensities are generalized (Eq. (9) and (10)). In the next step the generalized intensities from all durations are pooled together and the probability distribution function (PDF) is fitted to the pooled sample. The generalization of the annual maxima through the transformation with the two Koutsoyiannis parameters enables the assumption that the generalized annual maxima of dif-

ferent duration belong to the same population, thus they have the same distribution. This method of EVA is advantageous since the joint estimation with data pooled over all durations is robust against the sample uncertainties. Subsequently, a generalized extreme value distribution GEV is fitted to the sample of the generalized annual maximum intensities, using the method of L-moments. Equation (11) is the cumulative distribution function of the GEV distribution with $\mu$, $\sigma$ and $\gamma$ being the location, scale and the shape parameters of the distribution function. Variables i and I represent the annual maximum rainfall intensity

sample and its population respectively.

$$F(I < i; \mu, \sigma, \gamma) = exp\{-[1 + \gamma\frac{(i - \mu)}{\sigma}]^{-\frac{1}{\gamma}}\} \tag{11}$$

Here $\gamma$ is fixed to 0.1 as Koutsoyiannis (2004) suggests this value for a more robust estimation when dealing with return periods up to 100 yr. If the two remaining GEV parameters $\mu$ and $\sigma$ are known, the precipitation quantiles of the generalized intensities is calculated by (Eq. (12)), the quantiles are scaled back to actual intensities (Eq. (9)) and converted into rain $P_T$,

depth in mm.

$$i = \mu + \sigma[\frac{(-ln(F(I < i))^{-\gamma} - 1}{\gamma}] \tag{12}$$





## 3.2 Spatial Sampling Strategies and ADDF Curves

### 3.2.1 Single Location Sampling (SLS)

The Single Location Sampling scheme is depicted in Fig. 4 , A. The analyzed areas are concentric circles with different sizes

to the center of each study location presented in Fig. 1, B. The areal precipitation series are calculated for each of the areas and the annual maximum series (n = 20) is extracted for each duration and area size. Then using the Eq. (8) to (11) the quantiles are calculated for each area size and duration mentioned previously. A total number of values comprise one sample for $n_t$ = 20 yr and $n_{sp}$ = 1 single location.

$$n_{am}^{SLS} = n_t \times n_{sp} = 20 \times 1 = 20 \tag{13}$$

### 3.2.2 Multiple Location Sampling (MLS)

Figure 4 , B presents an example scheme of annual maximum sampling by MLS method. For each location, a larger area (r = 36 km) is considered as *study domain* where randomly distributed points (and areas) are chosen as sampling sites. The number

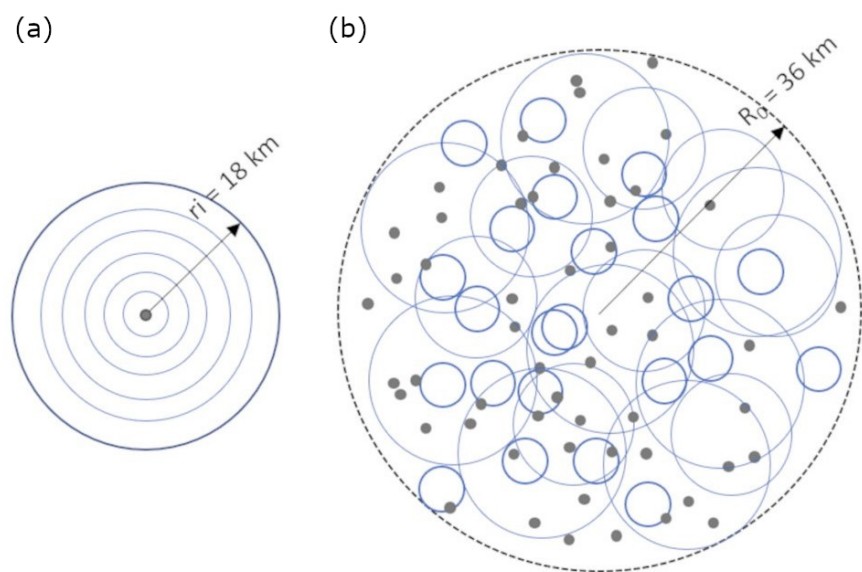

**Figure 4.** A: single location sampling of areal extremes (SLS), B: multiple location sampling (MLS, MLES and SLES). In both schemes the outer most circles are to the center of the study location. With SLS, areal precipitation of concentric circles with different sizes is calculated and then the annual maximum series is extracted from them. In MLS, MLES and SLES, a study region with a radius double the largest radius of interest is considered. Randomly distributed areas (and points) within this region are considered for sampling of the respective areal precipitation series and the extreme value analysis.





of the sampling sites $n_{sp}$, is chosen for each area so that the larger the area the smaller the number of samples taken: $n_{sp} =$ 500, 350, 200, 150, 100, 75 and 50 respectively for $a =$ 1, 12, 50, 113, 314, 615 and 1017 km$^2$. At each random site the time

series of areal precipitation is calculated and for all durations the annual maximum series and their time index, the time step at which the year's maximum occurs, are extracted. For each area size and duration, the annual maximum series from all the sampling sites are pooled together and the dependent events are removed. The removal of the dependent events is fulfilled such that, for each duration, the events which occur on the same day are filtered out and for each day with multiple events registered in the domain, only the event with the highest precipitation depth is kept in the sample. This results in a sample of pooled

annual maxima with

$$n_{am}^{MLS} = n_t \times n_{sp} \tag{14}$$

with

$$n_t = 20 \tag{15}$$

and

$$n_{sp} = 500, 350, 200, ..., 50. \tag{16}$$

sThe EVA is applied to the pooled annual maxima and the quantiles are calculated for each area size and duration.

### 3.2.3 Multiple Location Extreme Sampling (MLES)

MLES follows the same spatial sampling scheme and procedure as in MLS (Fig. 4, B) with the following additional step: after pooling the maxima together and removing the dependent events, only the 20 largest events for each duration are kept (n = 20):


$$am^{MLES} = [am_{(1)}^{MLS}, am_{(2)}^{MLS}, am_{(3)}^{MLS}, ..., n_{(20)}^{MLS}] \tag{17}$$

with a total sample size of

$$n_{am}^{MLES} = 20; n_t = 20. \tag{18}$$

### 3.2.4 Single Location Extreme Sampling (SLES)

The SLES method follows the same spatial sampling scheme as MLS and MLES, however the EVA is done in a different manner. In SLES, the sampling sites are chosen within the study domain and the annual maximum series are calculated for each sample as it was done in MLS and MLES. Then, instead of pooling the annual maxima together, the PDF is fitted to each sample of annual maxima separately using the Eq. (8) to (11) with

$$n_{am}^{SLES} = n_t = 20. \tag{19}$$





This step results in $n_{sp}$ quantile values for each duration, area size and return period. In the next step, for each duration, area size and return period, the maximum rainfall return depth is sought among the $n_{sp}$ values. Thus, the ADDF curves are presented by selecting the maximum value of the rainfall quantiles, for each duration, area size and return period within the study domain from

$$n_{sp} = 500, 350, 200, ..., 50 \tag{20}$$

where similar to MLS the value depends on the area size.

### 3.3   Spatial Order Problem Quantification

    Due to rainfall's intermittency, as the area increases the areal precipitation depth must decrease since the areal precipitation is the average of precipitation values within the considered area. In this study the ADDF curves are presented as sets of curves for different area sizes as shown in the results section. The expected decrease of areal precipitation with area must manifest as

a descending order of the curves with area. As presented and discussed vastly in the results and discussion section, our curves did not show the expected descending order over all durations. Instead, the decreasing order of curves changes to an increasing order after certain durations. Different examples of this issue are depicted in Fig. 5. As is evident the issue shows itself as crossings between the ADDF curves that belong to one study location. In order to make this problem more comprehensible, quantifiable measures are defined here. First for each set of ADDF curves at one location, the ***spatial order difference (SOD)***

is calculated for all durations:

$$SOD(d_i) = \frac{\sum_{k=1}^{n_A} |rank(Z_T(a_k, d_i)) - rank(Z_T(a_k, d_{i-1}))|}{n_A} \tag{21}$$

with $i = 2, ..., 10$ and $d_i = 5, 15, 30, 60, 120, 240, 480, 720, 1080$ and $1440$ min. $Z_T$ represents the areal rainfall quantile with the return period T and the $n_A = 10$ is the number of area sizes. The SOD is calculating the average rank difference of the areal curves, between one duration and the duration before that to determine whether or not the order of the curves have

changed from for duration to the next. The change in the curve ranks indicates a crossing of the curves between two durations.
    Then the following indicators are defined based on the *SOD*:

  – ***Number of Crossings (NC)*** represents the number of times that the order of the curves is not in descending order. This value is defined for one set of curves, for one location and return period.

$$NC = n[SOD(d_i) \mid SOD(d_i) \neq 0] \tag{22}$$

– ***Degree of Crossing (DC)*** indicates for the extent of the discrepancy between the observed spatial order of curves and the supposed descending order, in another words the complexity of the crossing in the curves. DC is defined as follows:

$$DC = \max_{d_i} SOD(d_i) \tag{23}$$





It is important to note that, in cases where the ADDF curves have more than one crossing ($NC > 1$) the DC refers to the crossing with the largest discrepancy from the correct order. More detailed explanations and examples are offered in the Discussion section of this paper.


- ***Duration of Crossing (CDur)*** is defined as the duration at which the crossing happens. Here as well, in cases where $NC > 1$, CDur is given for the largest crossing, namely the crossing with the maximum DC.

$$CDur = d_i \,|\, SOD(d_i) = \max_{di} SOD(d_i) = DC \tag{24}$$

## 4 Results

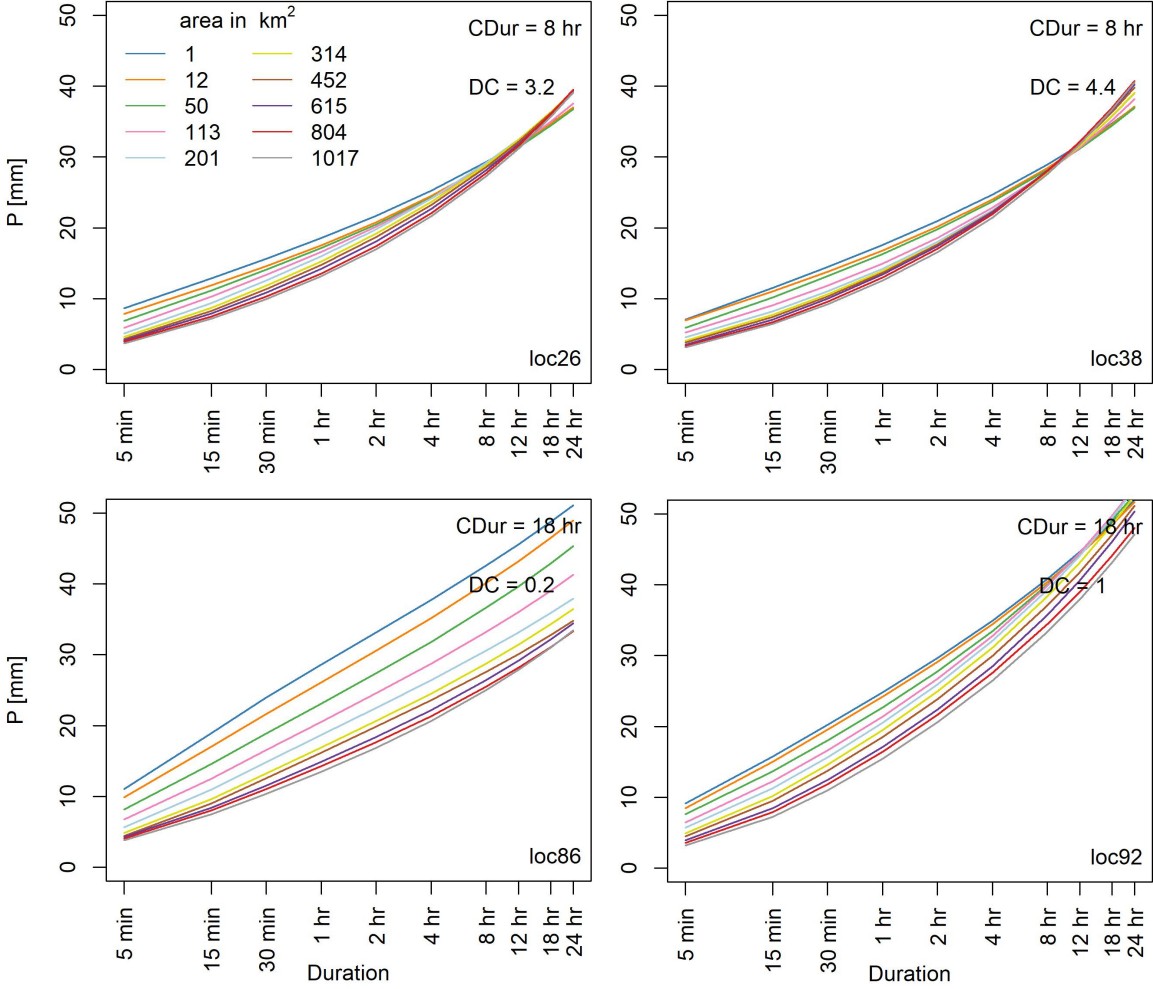

**Figure 5.** ADDF curves SLS, four different locations, T = 20 yr.





## 4.1 ADDF Curves - SLS

The ADDF curves resulting from the SLS method are presented in Fig. 5. Each curve set is depicted as the areal precipitation quantiles vs. duration for the return period of 20 yr, where the colors distinguish the areas. The panels of the figure show four examples from different study locations. The curve of the area with 1 $km^2$ represents one radar pixel, and is equivalent to a point. The areal extremes increase with duration and decrease with area for durations up to CDur = 8 hr in all examples. For longer durations than CDur the extremes increase with area at loc26 and loc38. At loc86 and loc92 this increase occurs partly only for a few areas. This change of behavior is the *spatial order problem* introduced earlier and will be discussed in detail in the following subsection.

### 4.1.1 Spatial Order Problem - *The Crossings*

The spatial order problem is depicted by *crossings* between the ADDF curves of different sizes. Since the areal precipitation decreases with increasing area, the areal curves should have a descending order with increasing area size. Here the expected descending order of curves changes (loc92) or becomes completely ascending (loc26 and loc38). This issue is called hereafter the *crossing* of the ADDF curves. The crossings are observed in 83% of analyzed locations where the majority of the crossings happen at durations longer than 8 hr (Fig. 6, B). The examples shown in Fig. 5 depict that the crossings have different degrees of complexity at different locations, as locations 26 and 38 has a larger DC than locations 86 and 92.

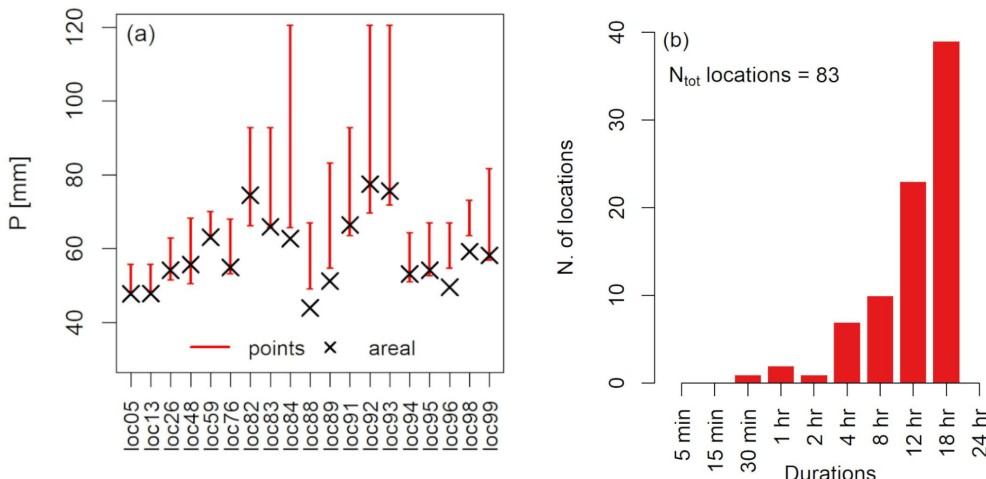

**Figure 6.** A: daily station precipitation extremes with T = 20 yr, from stations which fall within the study locations (r = 18 km). Red lines display the range of point precipitation, black symbols show the areal percentiles. B: Frequency distribution of the duration of maximum crossing for SLS-ADDF curves





### 4.1.2 Attribution of the Crossings

To investigate the source of this behaviour, the crossings were analyzed in relation to distance from radar center, topography and seasons. The ADDF curves from the maxima extracted from winter and summer time series separately, show that there are some differences in the behavior of crossings in summer and winter. Both figures 7 and 8 both show that in summer the number of locations with crossings is smaller than in winter, nevertheless in both cases, the crossings are observed in over 70% of the locations. There are locations which show no crossings, neither in summer nor in winter, and locations which do not show crossing in summer but in winter and vice versa. Another difference is observed in the duration and the degree of crossings (Fig. 7, A and B). In summer the majority of crossings occur at long durations, like 12 and 18 hr whereas in winter the crossings appear mostly at 4 hr, however, the number of locations with crossings at other durations is not insignificant. The ADDF curves of summer, have lower DC than the curves of winter (Fig. 7, C). Based on the distribution of the duration of crossing in space, no conclusive relational pattern is observed with topography nor with the distance from the radar center (Fig. 8).

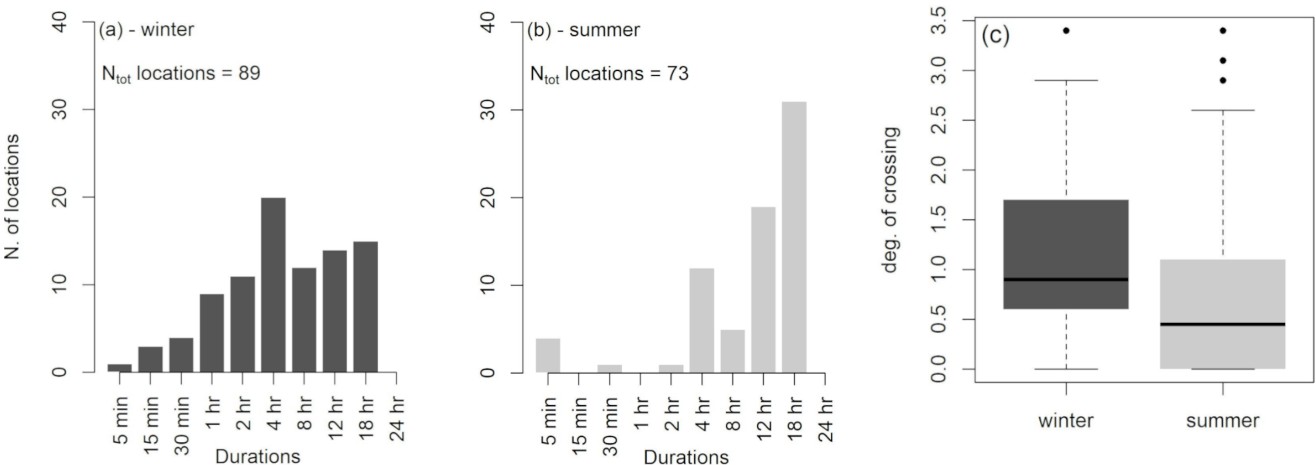

**Figure 7.** Frequency distribution of the duration of maximum crossing for SLS-ADDF curve, winter and summer

In order to investigate whether the crossings are an artefact of the radar data or their correction and merging processes, selected data from a dense daily rainfall station network are is used as a reference for this investigation. For that the areal and point extremes in a subset of study locations are examined. The comparison between the areal and point precipitation quantiles for T = 20 yr from the daily station data reveals that similar order relation problems occur with the station time series (Fig. 6, A). Out of 19 observed locations, 12 showed that the areal precipitation quantile for an area of 1017 km$^2$ is larger than the point precipitation at least for one station in the observed area. This observation is aligned with what the crossings in the radar-based ADDF curves show and confirms that the crossings are not an artefact of radar data processing.



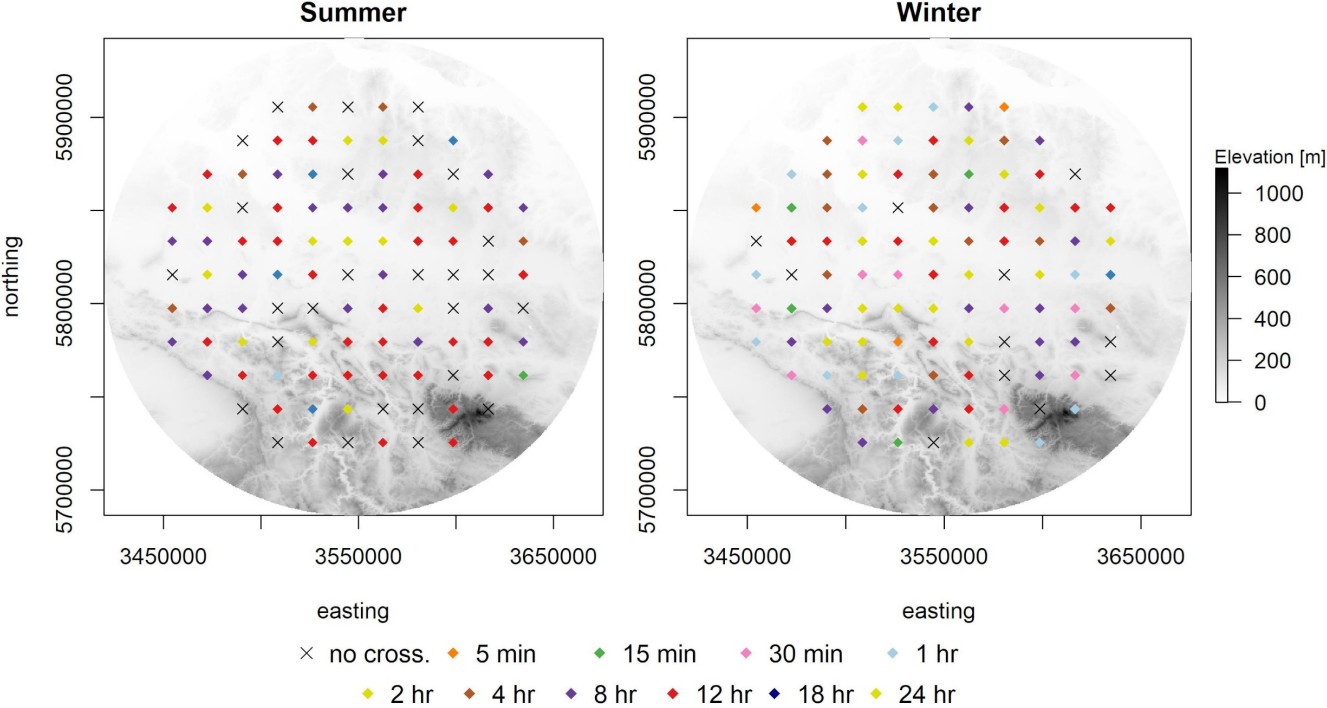

**Figure 8.** Relationship between the duration at which the crossings occur and elevation and distance from the radar center, in winter and summer, T = 20 yr.

## 4.2 ADDF Curves - Alternative Sampling Strategies

### 4.2.1 MLS

The MLS ADDF curves (Fig. 9) present higher values of precipitation quantiles in shorter durations, whereas for durations longer than 4 hr the values decrease and end up at approximately similar values to SLS ADDF curves. Moreover, with the MLS the crossings do not appear as frequently as SLS. In the rare cases in which crossings occur, the degree of crossing is small and it is often the curves of the larger areas crossing each other (Fig. 9, loc38 and loc92). Cases in which the point curve is crossed by an areal curve happen rarely ( < 5%). Figure 9 displays the ADDF curves for the same locations discussed in Fig. 5. As depicted, the crossings at locations 26 and 86 have vanished while at locations 86 and 92 only the curves of the three largest areas cross each other indicating the spatial order problem occurs only between three area sizes. The point precipitation (1 km$^2$) is still higher than all estimated areal precipitations.

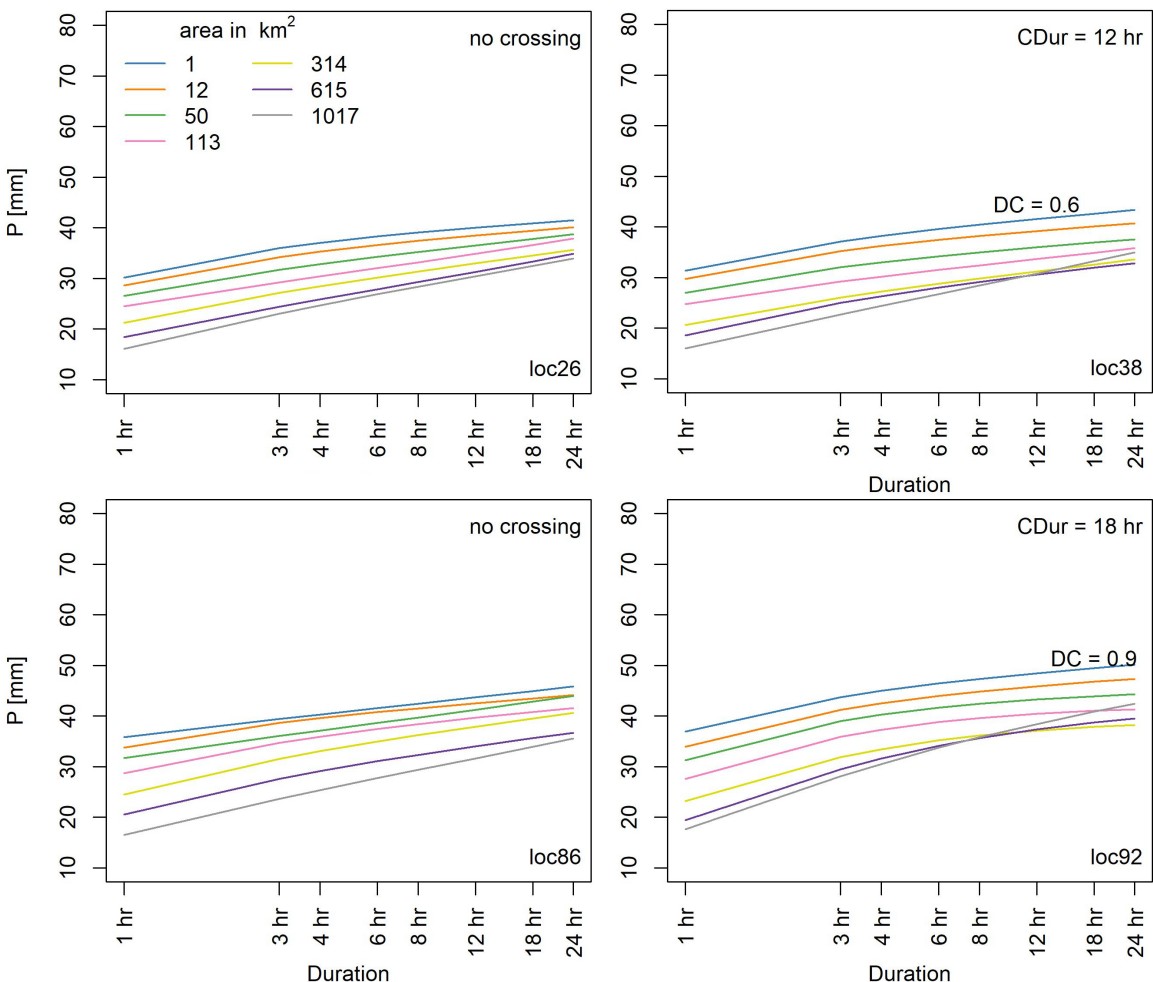

**Figure 9.** ADDF curves from the MLS, T = 20 yr. The results of MLS are calculated only for 7 of the 10 area sizes chosen at the beginning of the study, to lower the computational demand. Locations 26 and 86 show no crossings any more and the DC of locations 38 and 92 have reduced compared to SLS,

### 4.2.2 MLES

In MLES method, by sampling the 20 largest events within the study domain, the resulting ADDF curves present higher precipitation quantile values and almost no sign of crossings (only 2% of locations had a crossing, Fig. 12), indicating that by applying the MLES the precipitation quantiles decrease with increasing area. The curves for the four example locations show no crossings and evidently higher extreme levels than the results of SLS and MLS methods (Fig. 10). It is noteworthy



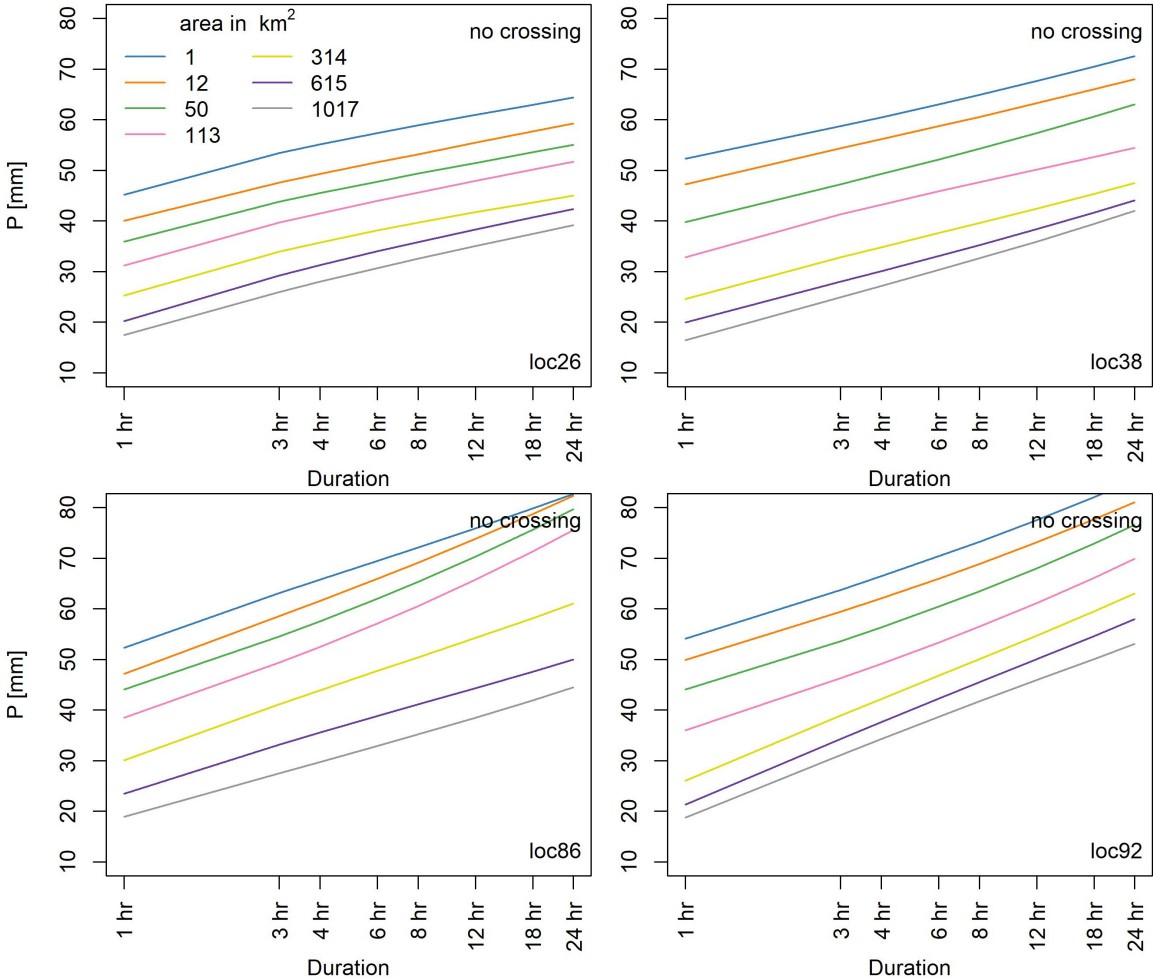

**Figure 10.** ADDF curves from the MLES, T = 20 yr. Compared to SLS and MLS results, these curves show no crossings at any of the example locations.

that MLES curves have a very low area-to-point (or ARF) ratio since the distance between the curves of different area sizes is
higher than in the other two methods.

### 4.2.3   SLES

Figure 11 shows the results using SLES for the four example locations. The ADDF curves resulted from SLES show quantile values generally higher than SLS results, where the difference is significant in curves of smaller areas; the larger the area becomes, the closer are the values of the SLES curve to the SLS. Compared to MLS results, the SLES shows higher precipitation
quantile values for longer durations, whereas for shorter durations the values of both methods approximate each other. Over





all, the MLES curves show higher values than SLES over all durations and area sizes. The frequency of crossings appearing in the resulting curves is higher in SLES than MLES methods but lower than in MLS and SLS (Fig. 12 , C). However the degree of crossing in the SLES crossings is considerably low like in MLES (Fig. 12 , B).

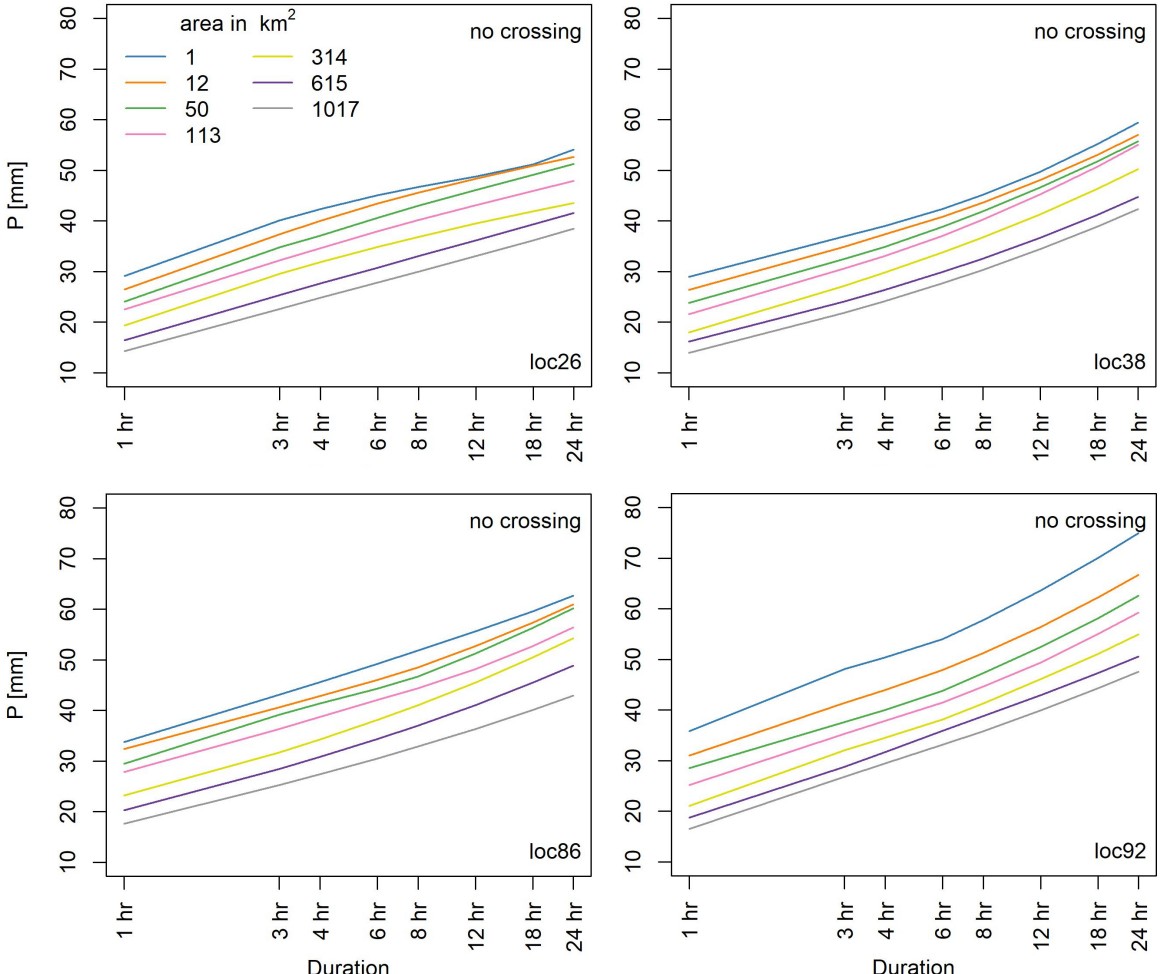

**Figure 11.** ADDF curves from the SLES, T = 20 yr. Compared to SLS, MLS and MLES results, these curves show no crossings at any of the example locations.

## 4.3   Crossings' Statistics

We compare the ADDF curves from the different sampling methods for durations of 1 to 24 hr and a return period of 20 yr (Fig. 12). To develop the alternative sampling methods only a subset of durations and areas (d = 1, 3, 4, 6, 8, 12, 18, and 24 hr; a = 1, 12, 50, 113, 314, 615 and 1017 km$^2$) are taken into consideration to reduce the computational demand of the analysis.





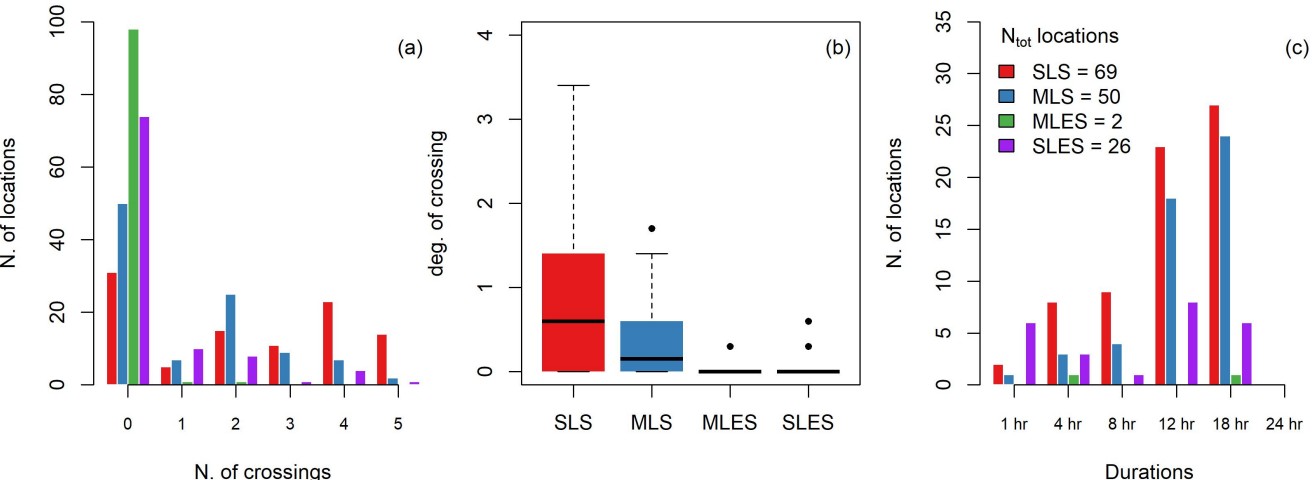

**Figure 12.** Crossing statistics for the three different strategies, A: frequency of locations with different number of crossings; B: distribution of the degree of maximum crossing among 100 locations for different sampling strategies and C: distribution of the duration of maximum crossing. All statistics are based on the ADDF curves with 20 yr return period.

Therefore the number of locations with crossings mentioned in this section is different from the number presented in Fig. 6. The choice of the return period of 20 yr in Fig. 12 is solely an example in presentation of results and has no significance over

other return periods. In SLS results, 69% of the locations present crossings and there are multiple cases with more than one crossing over all durations (Fig. 12, C), for example, there are locations with crossings of curves at five different durations. The statistics in Fig. 12 are based on the mentioned subset. The number of locations with crossings decreases in MLS, with 50% of curves showing crossings and the number of crossings within each set of curves decreasing. With MLES, the number of locations with crossings reduces to 2% where both locations have only one crossing in their respective set of curves. In SLES

this number increases to 26% which shows a decrease compared to SLS and MLS and an increase compared to MLES. Panel B, in Fig. 12, shows the variation of the degree of the crossing in three different scenarios, where clearly the degree of crossing decreases as we move from a single fixed location sample to a more spatially distributed sample of maxima.

## 5    Discussion

The ADDF curves are calculated for different area sizes at 100 locations using a radar-based rainfall data set and the observed

spatial sampling issue is investigated. Alternative sampling strategies of extremes are developed and compared. In this section we discuss the crossings issue, the reason behind it and what they mean. Further, we interpret the effect of the sampling methods on this issue and finally compare the point results of different methods with KOSTRA-DWD-2020 (Junghänel et al., 2022), the DWD's station-based design storm catalog for whole Germany.





## 5.1 The Spatial Order Problems and Attribution

As discussed in the previous sections, using a fixed-point/-area approach for sampling extremes leads to results in ADDF curves which are not consistent with the physical characteristics of rainfall extremes. This inconsistency appears as crossings in ADDF curves. A crossing in a set of ADDF curves, which indicates that from a certain duration (CDur) the extremes increase with area. The irregular and sporadic nature of rainfall leads to variations in intensity and duration over time and space. In smaller areas, intermittent and localized heavy rainfall events can significantly influence the average rainfall depth. As the area size increases, these localized events are spread out and averaged with regions experiencing less intense or no rainfall, leading to a lower overall average rainfall depth. In a fixed observation area with a single observation point at the center (or any point within the area), the largest event of the year for a certain duration may consist of multiple smaller cells drifting across the area in a scattered manner. It is likely that the rain cell with the largest intensity never passes over the observation point. In that case the areal precipitation will end up higher than or equal to the point precipitation. The same reasoning is valid for the areal precipitation of a fixed small observation area within a larger observation area and this is what leads to the crossing of the ADDF curves. In other words, where ever the crossings appear in ADDF curves, at the durations following the crossing, the point and/or all or some of the smaller areas missed the part of the annual maximum event which included the pixels with the highest precipitation depths. The findings from Bárdossy and Pegram (2018) support this interpretation.

The DC as a measure of how complex a crossing is can vary between 0 and 3.5 for the curve sets with seven area sizes (and from 0 to 5 with 10 area sizes). The larger the DC the more complex is the crossing. A higher DC, thus a more complex crossing, means more observation areas missed the extremes captured by larger areas, indicating that the areal annual maxima were sampled poorly compared to a case with a lower DC or no crossing at all. It is worth repeating here that, there are sets of ADDF curves, especially in SLS results, where the crossings appear at multiple durations. The DC and the CDur are only referring to the crossing with the highest degree of crossing within a set of ADDF curves.

To investigate this issue from the seasons (event types) perspective, the analysis was repeated on the extremes of winter and summer separately. In winter the dominant events belong to frontal systems which affect larger storm areas, whereas in summer the convective storms take the majority of the events, which are spatially concentrated and short in duration (Biondi et al., 2021). The winter ADDF curves show larger DC compared to summer (Fig. 7) and the crossings appear at almost all durations, whereas in summer the crossings happen predominantly around 12 - 18 hr. The frontal systems in winter, with larger spatial extent, have a higher spatial variability (Kim et al., 2019) and higher spatial variability can be the reason for a higher number of locations with crossings and more complex crossings when looking at winter extremes only. The same applies to summer, as with convective storms happening dominantly, longer temporal observation windows capture multiple convective cells which might be missed by the fixed point.

All the above points support the existence of the spatial order problem with traditional single location sampling. In order to reach an effective method for calculating areal extreme rainfall, this issue needs to be addressed.



## 5.2 Effect of the Sampling Method

To tackle the spatial sampling problem, alternative methods are offered. As is evident in figures 9, 10 and 11 the spatial order problem is addressed by all methods. In MLS and SLES the crossings reduce in number and degree and in MLES the crossings vanish almost completely with 98% of locations showing no crossings. This shows that by taking multiple areal samples across a domain surrounding the location of interest, the potentially missed extreme events will be covered by the observation window. Notably all three of the alternative sampling methods are more successful in capturing the real extremes. By pooling the extreme value distribution is assumed to be stationary, thus belonging to the same PDF. The MLES curves do not provide the extreme rainfall for a point or area but rather for any point or area within the study domain. Therefore, MLES cannot be used for design because it is overestimating the risk of extremes. Moreover it would lead to very small ARF values in practice, due to the very small point-to-area ratio of the curves mentioned in the results section. As an example, at loc92 (Fig. 10) the ARF for an area of 1017 $\text{km}^2$ would be roughly 0.3. The very low ARF values resulting from MLES curves, when applied to point precipitation quantiles which might already be underestimated would yield in significant underestimation in areal precipitation extreme and lead to high vulnerability of infrastructure build based on such calculation.

In SLES approach, although the number of the locations with the crossings is not insignificant, the degree of crossing is very low at all the locations which have the crossings. These cases are mostly where only two curves of 1 $\text{km}^2$ and 12 $\text{km}^2$ cross and in rare cases where the curves of 50 $\text{km}^2$ and 113 $\text{km}^2$ cross. The final ADDF curves are driven partly empirically as the maximum quantiles within are chosen for each duration and area size after the PDF fitting. In other words, the final resulting ADDF curves from the SLES method are not the direct outcome of the extreme value analysis, thus the curves have breaking points in them where the slope changes. Furthermore the SLES curvses present much higher point-to-area ratio and can yield in more realistic ARF values than MLES curves.

## 5.3 Comparison with KOSTRA

As there are no established methods for estimating spatial extremes directly from station rainfall data there is no reference at hand to validate the radar-based ADDF curves. Therefore a simple comparison is done on the point level between the 1 $\text{km}^2$ curves of the three sampling strategies and the extreme rainfall values from KOSTRA at the 100 study locations (Fig. 13). KOSTRA is the standard reference for station-based extreme rainfall in Germany, used by engineers and water authorities, and is provided as a grid covering the whole of Germany with 5 km resolution for different durations and return periods (Junghänel et al., 2022).

KOSTRA shows significantly higher values among all durations compared to SLS. This is due to the fact that the KOSTRA is based on a 60 yr station network time series whereas the radar product used for this study covers only 20 years. With a longer observation period there is a higher chance for the rain stations to capture larger extremes over time. Moreover, as observed in Fig. 3, panel D, although the radar product used in this study performs well at estimating rainfall extremes, the extremes are underestimated overall. This underestimation would propagate in spatial and temporal aggregation done in the extreme value





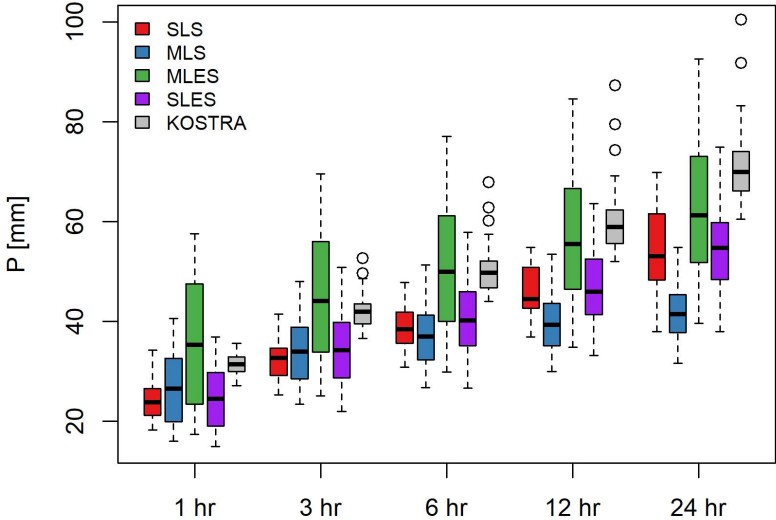

**Figure 13.** Extremes for T = 20yr at a point scale (a = 1 km$^2$) at the 100 studied locations. Comparison of different sampling strategy and KOSTRA

analysis and might be another reason for the discrepancy between the KOSTRA and SLS values. The disadvantage caused by radar data set's short length and the systematic underestimation affects all sampling methods.

Similarly, MLS shows smaller values than KOSTRA all durations. For durations longer than 6 hr the MLS extremes vary less with duration and they present smaller values than SLS. This is possibly because by sampling independent annual maxima from multiple sites in a domain of over 4000 km$^2$, we are adding annual maxima of different magnitudes to our final sample of maxima. Although the pooled sample includes the largest events of the domain, which is of interest in this method, the frequency of the smaller annual maxima than the most extreme events of the domain is increased in the final sample as well.

Thus although including the highest values within the domain is favorable to our goal, the addition of a high number of medium level extremes affects the parameters of the PDF drastically. The presence of a larger number of smaller annual maxima in the sample leads to less variable quantiles among durations and a biased sample which leads to significant underestimation of the extremes. Figure 14 as well shows the low variability of the MLS ADDF curves over durations and the underestimation easily detectable in panel b as the MLS extreme values of longer durations are smaller than the extremes of SLS.

MLES values have a median which approximates the KOSTRA results but generally a much wider inter-quartile range as compared to KOSTRA. For sub-daily durations, the MLES sampling leads to higher extremes than KOSTRA and for daily events KOSTRA has higher values than MLES. This can be explained by the longer duration and of the events and observation period. A longer temporal window has better chances of capturing the storm traces. This, in combination with long time series of observation in KOSTRA, can explain KOSTRA showing higher extremes than MLES which is based on only 20 years of

data.



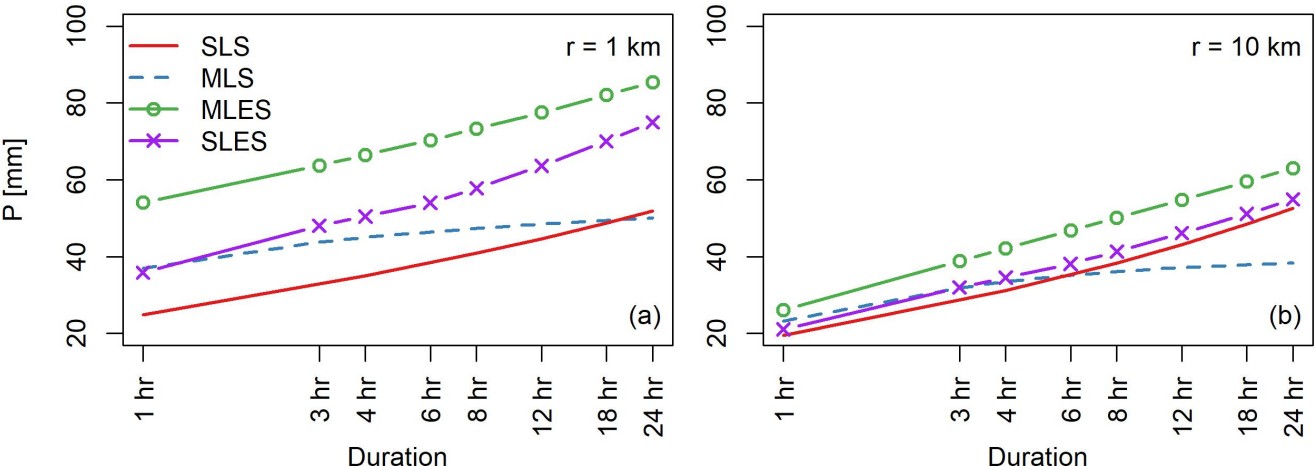

**Figure 14.** Comparison of ADDF curve resulted from different sampling strategies. Results from one location, T = 20 yr, areas with r = 1 and 10 km, A and B respectively

SLES point extremes have higher values than SLS in all durations, and than MLS for durations longer than 3 hr. Events with 1 hr and 3 hr durations are approximately the same in the MLS and SLES methods. Compared to KOSTRA, SLES shows lower values, however the discrepancy between SLES and KOSTRA is smaller than the difference between MLS or SLS and KOSTRA.

Among the three tested sampling strategies, we recommend the SLES method for estimating the representative point extreme or calculating the area extreme of a catchment directly, as it presents the maximum precipitation quantile for the region. Moreover, for the purpose of calculating ARF it is a more suitable method, especially for a case like Germany, where the design point rainfall catalog is solely based on station data.

## 6 Conclusions

In this study we aimed to develop a method for moving from a point approach to more spatial approach in design rainfall estimation by utilizing a radar-based product. The initial attempts for calculating the ADDF curves showed an implausible pattern: the crossing of the curves and the increase of rainfall quantiles with increasing area. To validate the truthfulness of the observed behavior, different hypotheses were tested. A comparison with the EVA results of a dense daily station network revealed the same behavior which lead the authors to question the sampling methods. As a solution to this problem, three

innovative alternative sampling methods are presented and discussed.

The main findings of this paper are:

- By observing annual precipitation maxima through fixed points and areas, there is a risk that the observation window may not capture the rain cells with highest rainfall depth as the event might bypass the observation window in space.





Missing the highest depths of the annual maximum events manifests as crossings of ADDF curves representing different area sizes and depicts an increase of precipitation depth with area size. The crossings in ADDF curves show that the point sample and the samples of the smaller areas, have missed the rain cells with high rain depths which were captured by the samples of larger areas.

– To tackle this issue multiple-location sampling methods are presented in which samples of annual maxima are taken from multiple points and areas randomly distributed within a study domain, encompassing the location of interest. The MLS takes the pooled series of annual maxima from all sampling sites, whereas MLES takes the 20 largest events of the pooled sample from the region. The SLES looks at the extremes of each sampling site separately and takes the highest quantile values within the region for each duration and area size.

– The MLES method proved to be the most successful among the three in removing the crossings from the curves, however it would lead to significant overestimation of point extremes and low ARF values due to the large distances between the curves. Currently available point design rainfall catalogs like KOSTRA are based on conventional methods, thus might underestimate the reality of the point extremes. Therefore, MLES cannot offer a suitable practical solution because applying low ARF to underestimated point extremes might lead to significant underestimations.

– The MLS method shows low values, partly even lower than SLS due to the presence of a high number of medium level annual maximum rainfall depth in its sample, and this method is not as successful as MLES and SLES in removing the crossings.

– The SLES method is the method that would be recommended here as a starting point for a practical solution to this problem, since it reduces the number of location with crossings from 69% to 26% and although it does not reduce the locations as well as MLES does, the remaining location with crossings in SLES method show only insignificant degree of crossings. Moreover, the ARF values which would result from this method are more realistic compared to MLES.

– The discrepancy of the point extremes between KOSTRA values and the results of the alternative sampling methods is due to a) the short length of the observed time series by the radar product (20 yr) compared to the stations data sets used for KOSTRA (60 yr), b) the minimal level of underestimation of the rainfall extremes by the radar merged product. Both of these shortcomings affect the three alternative sampling methods equally. Further developing the method of radar data correction and merging, as well as using advanced statistical approaches would help compensate the short length of radar records. However further investigations are needed which do not fit in the scope of this study.

The issue of extreme's underestimation by conventional methods has been rarely discussed in the hydrological community (Bárdossy and Pegram, 2018). To our knowledge this paper is the first study investigating this issue in details and more importantly offering a solution. The authors are convinced that moving from a point understanding of the rainfall extremes to a spatial understanding is crucial and that the SLES method is a strong starting point for developing robust solutions for engineering practice.





*Code availability.*  All R codes can be provided by the corresponding authors upon request.

*Author contributions.*  UH acquired the supervision and funding for this research. Both authors contributed to the study conception, design, and methodology. GG undertook the tasks of programming, data collection, result derivation, and interpretation. The original draft was prepared by GG and subsequently revised by UH.

*Competing interests.*  The authors declare that they have no conflict of interest.

*Acknowledgements.*  The authors express their thanks of gratitude to the German Federal Ministry of Education as well as the Research and the State Ministry of Agriculture and Environment of Mecklenburg-Western Pomerania for the financial support of this research project. Moreover we would like thank the DWA working group, especially Winfried Willems for discussing the sampling methods.



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
