# Peer review of "Estimation of radar-based Area-Depth-Duration-Frequency curves with special focus on spatial sampling problems"

_Hydrology and Earth System Sciences, 2024_

## Referee Comment (RC1)

The paper presents an important and original contribution to the sampling problem in estimation of Area-Depth-Duration curves. These models are important in hazard assessments among other applications. Overall, I enjoy reading the paper, it's clear, easy to follow and contains very relevant discussions. I have some minor comments that I believe need some clarifications, after which I recommend the paper for acceptance.

- In the methodology, the authors fitted the IDF of Koustoyiaannis et. al. (1998) for each area separately. However, there are IDAF formulations that links all the data for different durations and different area together and fit one IDAF formulation. An advantage of this is that a constrain is already implemented in the formulation to ensure that the intensities decrease with area. Example is the IDAF formulation of De Michelle. Did the authors consider this option?
- In the multiple location sampling, a random sampling is done. I expect that each time the sampling is repeated, a different set of locations will be selected. Would this affect the result? Have you considered using a moving window in space to capture all possibilities?
- I find it surprising that the spatial crossing does not show any pattern as a function of the topography. For instance, Melese et al. (2019) and Haruna et al. (2024) observed this behavior to depend on orography, for instance the location of the study pixel on the windward or leeward side. Could you comment on this? Would it be possible to apply the different sampling strategies to a pixel on the leeside or wind side of the mountain and to see the effect on the curves?

    Melese, V., Blanchet, J., and Creutin, J.-D.: A Regional Scale–Invariant Extreme Value Model of Rainfall Intensity–Duration–Area– Frequency Relationships, Water Resources Research, 55, 5539–5558, https://doi.org/10.1029/2018WR024368, 2019.

    Haruna, Abubakar, Juliette Blanchet, and Anne-Catherine Favre. "Estimation of Intensity-Duration-Area-Frequency Relationships Based on the Full Range of Non-Zero Precipitation From Radar-Reanalysis Data." *Water Resources Research* 60.2 (2024): e2023WR035902.

    Rosin, T., Marra, F., and Morin, E.: Exploring patterns in precipitation intensity–duration–area–frequency relationships using weather radar data, Hydrol. Earth Syst. Sci., 28, 3549–3566, https://doi.org/10.5194/hess-28-3549-2024, 2024.

- I was expecting "Summer" to have more locations with crossing, compared to winter. Since summer events are convective and tends to be isolated, while winter are stratiform and tends to cover a larger area. Could you comment? Furthermore (in Line 375), I expected frontal systems to exhibit less spatial variability compared to convective events. Since frontal storms are driven by large-scale interactions between air masses rather than localized convective processes, the intensity of precipitation and weather conditions tends to be more uniform compared to convective storms. Could you comment?
- The authors compare the quantiles from the various sampling strategies to those from KOSTRA. Due to the inherent differences between the two, I don't understand how KOSTRA values could serve as benchmark for preferring one method over the other. Should the best method agree with KOSTRA values? Why?
- Is there any motivation for the choice of the different areas and durations, and more precisely the upper bounds?
- Do you observe the same "crossing" based on simple exploratory analysis of the annual maxima series (without fitting GEV).
- Line 93: I don't understand the sentence "The region has been observed for the time period of 2000 to 2019,…" Could you rephease.
- Line 101. Do you mean "final merged radar data"?
- Line 155: Eq 7 instead of 6

- Line 194: "…actual intensities". Do you mean "intensities with durations d"?
- Line 207. The largest area (R=36 km), any justification for this choice? Would it affect the result?
- Line 415: "…..smaller values than KOSTRA all durations." - > ".. for all durations"
- Figures
  - In Figure 1, it is difficult to contextualize the location of the study area with respect to the map of Germany, Would it be possible to add a locator map?
  - What are the grey colored points that are randomly located in Figure 4b. They seem to be independent of the circles.
  - Figure 4: Add that the circles are colored according to the radius(area)
  - Caption of Figure 4, the phrase "In both schemes the outer most circles are to the center of the study location." Seems not complete.
  - Figure 5 and 9. The choice of the color palette (seems to be discrete/qualitative) makes it difficult to track the changes of the quantiles as functions of the area. Since the area are increasing, a "sequential" or "diverging" palette would be better (Eg. Melese et al. (2019)).
  - Figure 6 B is mentioned earlier (Line 284) than 6 A (Line 300). Consider switching the two?? (Moving 6 B to the right)
- Equations
  - Check equation 11 carefully, a power is missing
  - The transition from Eq 8 to 9 is not very clear. For example, the index $a$ is suddenly dropped. I suggest rewriting it for clarity.
  - In Eq 14, since dependent events are removed, then $n_{am}^{MLS} \leq n_t \times n_{sp}$. Am I correct?

---

## Author Comment (AC1)

Dear reviewer,
Thank you very much for your thorough review. I have tried to reply your comments as clearly and derailed as possible. I am happy to discuss further with you any ambiguities that might still be there.
Kind regards,
Golbarg

• In the methodology, the authors fitted the IDF of Koustoyiaannis et. al. (1998) for each area separately. However, there are IDAF formulations that links all the data for different durations and different area together and fit one IDAF formulation. An advantage of this is that a constrain is already implemented in the formulation to ensure that the intensities decrease with area. Example is the IDAF formulation of De Michelle. Did the authors consider this option?

GG: This is a good point. We are aware of the analytical formulations of the IDAF/AIDF curves which include not only duration but area as well and our final goal is also to provide a formulation like this. However, the inconsistencies in IDAF/AIDF curves (crossings) are so heavy that we think it is better first to reduce the crossings based on their real cause as much as possible. For that we are looking for an optimal sampling approach. In a second step we would like to find an analytical formulation which includes the area to avoids the remaining order relation problems. This is however out of the scope of this paper.

• In the multiple location sampling, a random sampling is done. I expect that each time the sampling is repeated, a different set of locations will be selected. Would this affect the result? Have you considered using a moving window in space to capture all possibilities?

GG: Well yes, when repeating the study, the random samples of points and areas will be different. However, our samples are taken so large that it ensures all the events within the study region around each location are covered. While removing the dependencies among pooled events there are always a considerable number of events being removed. This is an indicator that the samples are covering all possibilities.

• I find it surprising that the spatial crossing does not show any pattern as a function of the topography. For instance, Melese et al. (2019) and Haruna et al. (2024) observed this behavior to depend on orography, for instance the location of the study pixel on the windward or leeward side. Could you comment on this? Would it be possible to apply the different sampling strategies to a pixel on the leeside or wind side of the mountain and to see the effect on the curves?

Melese, V., Blanchet, J., and Creutin, J.-D.: A Regional Scale–Invariant Extreme Value Model of Rainfall Intensity–Duration–Area–Frequency Relationships, Water Resources Research, 55, 5539–5558, https://doi.org/10.1029/2018WR024368, 2019.
Haruna, Abubakar, Juliette Blanchet, and Anne-Catherine Favre. "Estimation of Intensity-Duration-Area-Frequency Relationships Based on the Full Range of Non-Zero Precipitation from Radar-Reanalysis Data." *Water Resources Research* 60.2 (2024): e2023WR035902.
Rosin, T., Marra, F., and Morin, E.: Exploring patterns in precipitation intensity–duration–area–frequency relationships using weather radar data, Hydrol. Earth Syst. Sci., 28, 3549–3566, https://doi.org/10.5194/hess-28-3549-2024, 2024.

GG: Yes, this was a surprise for us as well. We are aware of the studies you mentioned and they are also cited in the manuscript. However, it is important to consider that our study area is pretty flat and there is just a small portion of it which has an overlap with the Harz mountain and due to the removal of the data at the edges of the study area to avoid including missing values and pixels

which are heavily affected by the radar errors it is not much information at hand at such locations. It is definitely interesting to investigate this further with data from other regions and climates.

• I was expecting "Summer" to have more locations with crossing, compared to winter. Since summer events are convective and tends to be isolated, while winter are stratiform and tends to cover a larger area. Could you comment? Furthermore (in Line 375), I expected frontal systems to exhibit less spatial variability compared to convective events. Since frontal storms are driven by large-scale interactions between air masses rather than localized convective processes, the intensity of precipitation and weather conditions tends to be more uniform compared to convective storms. Could you comment?

GG: The crossings appear mainly for longer durations, 4hr in winter and 12-18hr in summer. This means they are not coming from small scale thunderstorms but from events with longer durations. To produce crossing these events need to be quite heterogenous in space. So, one explanation could be that these are events where frontal systems are overlaid with convective parts. These events can occur in winter and in summer and it can be assumed that the frequency of those mixed events is increasing with global warming. There was also a misinterpretation form our side based on the cited study Kim et al 2019. We rewrote that paragraph.

• The authors compare the quantiles from the various sampling strategies to those from KOSTRA. Due to the inherent differences between the two, I don't understand how KOSTRA values could serve as benchmark for preferring one method over the other. Should the best method agree with KOSTRA values? Why?

GG: Of course, only the point-related IDF curves (a = 1 km$^2$) can be compared between KOSTRA and the radar product. This is what is done here. The radar product is a merged data set considering station data as truth and so is supposed to resample the station statistic at a point quite well (see Fig. 3). For both products KOSTRA and the merged radar data the extreme value analyses is done using the same approach. In KOSTRA an additional regionalization step is included which makes things more uncertain than for a direct point estimation. On the other hand, KOSTRA is based on 60 years data and the radar analyses only on 20 years, which is probably the largest difference between the two. Altogether we think the point radar IDF curves should approach the KOSTRA IDF curves from below due to the difference in observation length.

• Is there any motivation for the choice of the different areas and durations, and more precisely the upper bounds?

GG: nothing specific. R = 18 km gives an area of roughly 1000 km$^2$ and area sizes ranging up to 1000 km$^2$ are representative of the catchments in the studies area. We have to assume stationarity within the area, which is as less guaranteed as larger the area becomes.

• Do you observe the same "crossing" based on simple exploratory analysis of the annual maxima series (without fitting GEV).

GG: yes, we do, even more complex crossings.

• Line 93: I don't understand the sentence "The region has been observed for the time period of 2000 to 2019…" Could you rephrase.

GG: done

• Line 101. Do you mean "final merged radar data"?

GG: yes, corrected

• Line 155: Eq 7 instead of 6

GG: actually, it refers to both of them, corrected

• Line 194: "…actual intensities". Do you mean "intensities with durations d"?

GG: I don't get your question exactly but I tried to make the sentence a bit clearer.

• Line 207. The largest area (R=36 km), any justification for this choice? Would it affect the result?

GG: the largest radius we worked with in the base method SLS was 18 km because it gives an area of roughly 1000 km$^2$. And doubling that to 36 km was to ensure we can take enough samples from the surrounding area. No specific reason for these numbers in first place. The analysis was done for whole Germany with the station dataset used in section 4.1.2. and we saw the same pattern for an area as big as Germany and station data.

• Line 415: "...smaller values than KOSTRA all durations." - > "... for all durations"

GG: corrected

• Figures

o In Figure 1, it is difficult to contextualize the location of the study area with respect to the map of Germany, would it be possible to add a locator map?

o What are the grey colored points that are randomly located in Figure 4b. They seem to be independent of the circles.

o Figure 4: Add that the circles are colored according to the radius(area)

o Caption of Figure 4, the phrase "In both schemes the outer most circles are to the center of the study location." Seems not complete.

GG: Fig 4 adapted

o Figure 5 and 9. The choice of the color palette (seems to be discrete/qualitative) makes it difficult to track the changes of the quantiles as functions of the area. Since the area are increasing, a "sequential" or "diverging" palette would be better (Eg. Melese et al. (2019)).

GG: your concern is valid. The initial color palette was rainbow which showed the areal dimension pretty well, but that palette is not colorblind friendly and that is one of the requirements of this journal. We tried so many different palettes, sequential etc. this was at the end the best compromise.

o Figure 6 B is mentioned earlier (Line 284) than 6 A (Line 300). Consider switching the two?? (Moving 6 B to the right)

GG: corrected

• Equations

o Check equation 11 carefully, a power is missing

GG: corrected

o the transition from Eq 8 to 9 is not very clear. For example, the index $a$ is suddenly dropped. I suggest rewriting it for clarity.

GG: corrected. Not the equation, but how the text leads up to it. Hopefully it is easier to understand now?

o In Eq 14, since dependent events are removed, then $n_{amMLS} \leq n_t \times n_{sp}$. Am I correct?

GG: correct, thanks

---

## Author Comment (AC2)

Main points:

1.  Underlying assumptions.

    - The authors seem to start from the assumption that increasing DDF with area are "implausible" (e.g. see the introduction or line 242: "as the area increases the areal precipitation depth must decrease"). While this is generally what one expects from a statistical perspective, it strictly holds only under spatial stationarity. There exist situations in which this can be not the case - e.g. see Mélèse et al. 2019 (https://agupubs.onlinelibrary.wiley.com/doi/10.1029/2018WR024368). Results in fig. 8 do not show any pattern, but I think some words on this aspect would be useful.

    GG: Yes, stationarity is assumed here within the circle areas, which are restricted in radius to be <= 18 km for single location sampling and <= 36 km for multiple location sampling, respectively. In addition, the cases with ARF > 1 mentioned in Melese et al. 2019 are now pointed out in the discussion section 5.1. last lines of the first paragraph.

    - The second assumption seems to be that the crossings are mainly due to sampling issues. This seems however in contradiction with some of the results: in lines 288-289, it is observed that "in summer the number of locations with crossings is smaller than in winter". Spatial sampling issues are expected to be more important for events with small space-time scales, such as summer events that are more often convective in nature, rather than winter events. How is this reconciled with the second assumption above?

    GG: Yes. The crossings appear mainly for longer durations, 4hr in winter and 12-18hr in summer. This means they are not coming from small scale thunderstorms but from events with longer durations. To produce crossing these events need to be quite heterogenous in space. So, one explanation could be that these are events where frontal systems are overlaid with convective parts. These events can occur in winter and in summer and it can be assumed that the frequency of those mixed events is increasing with global warming. Maybe our discussion was not clear enough. Also, one of the studies mentioned in the discussion was interpreted somewhat wrong by us. We rewrote the discussion part on the seasons. The paragraph starting at line 370 is revised as below:

    *To investigate this issue from the seasons (event types) perspective, the analysis was repeated on the extremes of winter and summer separately. In winter the dominant events belong to frontal systems which affect larger storm areas, whereas in summer the convective storms take the majority of the events, which are spatially concentrated and short in duration (Biondi et al., 2021). The winter ADDF curves show larger DC compared to summer (Fig. 7) and the crossings appear at almost all durations, whereas in summer the crossings happen predominantly around 12 - 18 hr. During winter frontal events occur more predominantly. Such events are characterized by their larger spatial and temporal extent. If the event is drifting over a part of the observing circles and the most intense cells are not passing over the center point and smaller areas, a higher number of areas captured a higher areal precipitation than the point in the center or areas of smaller sizes Which leads to a higher DC. The same applies to the frequency distribution of the CDur. In winter the crossings appear with a considerable frequency at most of the studied durations, ranging from 1 hours to 24 hours. Frontal events' longer temporal spans allow shorter observation windows to capture the same prolonged event. On the contrary summer is associated with convective events with shorter durations and smaller areas. A crossing with convective events happens when a) one event is*

*captured partially by the observing areas or b) multiple events drift over the circles. In both scenarios the events drift over the observing windows so that the areas are not completely covered by them and the cells with storm centers with the highest intensities passes through the area closer to the edges than center. Since the events are smaller specifically in scenario a, the DC is smaller since it is likely that not all the observing areas capture the event. Scenario b happens at the longer durations, since the convective events last shorter, when the temporal observation window is long, it is more probable to capture multiple events as in one observation. The number of locations with crossings is slightly lower in summer because the fixed spatial sampling and smaller area coverage of convective events increase the likelihood of missing some events entirely.*

- In lines 80-081, it is claimed that "we investigate the spatial order relation problems, appearing as crossings in ADDF curves, which lead to missing information in areal rainfall extreme value analysis and underestimation of design storms". This sentence seems to assume that missing a storm leads to underestimation of the statistics. From a population perspective, "missing" a particular storm is part of the local climatology (the event did not hit the place of interest). The problem arises when the sample at hand is limited, and the missing may be considered a statistical outlier. This leads to the question: how much is the problem related to use of a block maxima approach and how much is it general?

GG: Maybe this is an issue of language and the sentence should be reformulated to convey our message. We surely understand that from a statistical point of view we do not capture ALL the storms. However, now with high-resolution radar data and trying such spatial sampling methods, we see that for events of longer duration and larger spatial extent, the captured annual maxima by our conventional observational methods are not representative of the truth of the rainfall as a spatial phenomenon. In our opinion the sample at hand is limited in so If these crossings were caused by outlier storms, they would not have appeared in the majority of the studied locations. The observation that the crossings arose in 83% was an indication of the limitation of the samples. In addition, from a spatial point of view in the block maxima approach, the largest observed values are selected but we still fail to capture some higher events, which do not occur in the center of the circle. When we would use the peak over threshold (or partial series or metastatistical approach), we would collect more events but would still miss the same extremes occurring not in the center.

Sampling methods.

I had some concerns with the MLES and SLES sampling methods, in which a maximum in space of maxima in time is extracted, because of the different sample size at different durations. Intuitively, this would lead to higher chances of having a large value in the small scales (more samples). I am proven wrong by figure 13, at least for the MLES sampling method. I'd be happy to see some discussion on this aspect.

GG: I am not sure if you mean MLES or SLES. In figure 13 the MLES (green) at all durations is showing a median considerably higher than the median of the SLS (the base method – red). Which does make sense because in the MLES method the PDF is fitted to the 20 largest events in the region all together. In SLES on the other hand we look at nsp randomly distributed points (and areas) within the region. Each point (or area) has an AMS with 20 values. The PDF is fitted to the

20 values at each of the random points (or areas). Then within the region among the nsp points (or areas) the quantiles with the maximum value, for each duration, area and return period is picked. In SLES we basically put the ADDF together in an empirical manner.

2. Quantitative accuracy.

Comparing fig 5 with fig 9, 10, and 11 shows huge differences between the quantities estimated in several locations. For example, in loc26, the 1 km2 scale at 5 minutes changes from 10 mm to 30 mm in the MLS and SLES (3x more). This is even larger (~4x more) in loc92 and even more for the MLES method in fig 10 (almost 6x more).  How can all these estimates make sense? Which ones make more sense from a quantitative perspective? The answer to these questions is only provided in the discussion (section 5.3). In my view, this should be the first comparison to be shown across the different sampling methods (in the results and before figures 9—11. For the same reasons, I suggest to include KOSTRA estimates in Fig 14a (1 km2).

GG: We agree with your point and these are very important questions. But adding the answer to these question in the results section and before the figures 9-11 will disturb the flow of the paper in our view. Therefore, we added a few sentences in the results sections respective to each of the alternative methods and also before starting the next topic we refer the readers to section 5.3. Figure 14a: adapted

3. Uncertainty.

- Figure 13 shows that MLES seems the optimal sampling method on average, although a very large variability in the results is observed. This could be due to a larger uncertainty related to this method.

GG: Yes, MLES has the highest uncertainty among all methods discussed. This sampling method uses only the 20 largest values and a small sample. So, the estimation is sensible to outliers or high values.

- There is no quantification of estimation uncertainty with any of the methods. This is an essential component of DDF and ADDF curves and should accompany the design values that are provided.

GG: Yes, there is no direct assessment of uncertainty, which of course needs to be done when the estimation technique is evaluated. However, our aim here was to focus first on the introduction of the crossing problem and potential alternative spatial sampling approaches and not yet on the design values.

- Uncertainty could be one of the reasons behind the observed crossings. For example, it could be that several of the crossing lines are within each other's uncertainty, thereby indicating that in some occasions, crossing may just be due to uncertainty (e.g. loc26 in figure 9 could well be the case). This for example was shown by Rosin et al. 2024 (cited in the manuscript). This option would be supported by the absence of clear patterns in Fig 8.

GG: We don't believe that the crossings for almost every location (83 from 100) in SLS are a result of the uncertainty. The crossings are a systematic error. They show systematic underestimation of point or small area values compared to larger area means. However, the few remaining crossings in the multiple sampling methods might be due to uncertainty (e.g. from Fig. 9) but probably still more due to remaining sampling problems.

- Should this be part of the reasons behind the crossings, it would be natural to ask in what proportion this may be related to the used method (here, Koutsoyiannis 1998). Could another method that already prescribes no-crossings be preferred? There are several used in ARF estimation that can be extended to the duration-area problem - e.g., De Michele et al 2001 (https://agupubs.onlinelibrary.wiley.com/doi/10.1029/2001WR000346)

GG: Another EVA method was applied initially which lead to even more complex crossings, also partly with multiple crossings in one set of curves. The tested method was the conventionally used EVA method in the German design storm estimation regulations (DWA-531). We are aware of the analytical formulations of the IDAF/AIDF curves, which account for both duration and area, and our ultimate goal is to develop a similar formulation. However, the significant inconsistencies in IDAF/AIDF curves (such as crossings) suggest that it would be more effective to first minimize these crossings by addressing their underlying causes. To achieve this, we are exploring an optimal sampling approach. As a subsequent step, we intend to establish an analytical formulation incorporating area to resolve any remaining order relation issues, though this goes beyond the scope of the present paper.

4. Previous literature.

- Line 85: "To our knowledge, there are no studies investigating the spatial order problem in detail and offering new sampling methods." I believe something is out there, for example Goudenhoofdt et al. 2017 (https://doi.org/10.5194/hess-21-5385-2017) and Poschmann et al 2021 (https://nhess.copernicus.org/articles/21/1195/2021/). I invite the authors to discuss their method in comparison to the ones proposed here.

GG: As far as I understand these two studies go in very different directions than this manuscript is taking. Goudenhoofdt et al. 2017 focuses on the application of radar QPE to estimate extreme precipitation at point scale and the regional frequency analysis. The former is close to what we have done in the point data validation section but that is merely the beginning step for starting the spatial analysis. The RFA of their study, is not quite comparable to our research since a major point in our attempts is to incorporate the spatial scale of the rainfall into the extreme value analysis, whereas Goudenhoofdt et al. 2017 do not take areal samples. In case of Pöschmann et al. 2021 as well, they are focusing on the temporal scaling relationship of rainfall extremes in Germany. Had they focused on the spatial scaling relationship, it could have been of interest as previous literature for this manuscript.

This is based on my understanding of these two studies, I would gladly discuss this further with you if there is a misunderstanding from my side on these two studies.

- Some parts of the manuscript (fig 6 and lines 297-303) reminded me of a paper by Peleg et al. 2018 (http://doi.org/10.1016/j.jhydrol.2016.05.033) in which small scale variability of extremes was quantified and compared in terms of the resulting areal estimate. The discussion in said paper may be relevant to the interpretation of this study.

GG: Thanks for the recommendation. The paper indeed has some relevance to our interpretations. Specifically, regarding the interpretation of the point validation in data section and why the final merged radar data is underestimating. Also, at in the discussion section it can be mentioned as one of the potential reasons why the radar estimates at point scale are lower than the KOSTRA estimates. This will be integrated within the text at the relevant spots.

5. Codes availability.

Given HESS policy on the matter, I encourage the authors to submit the codes to an open repository for public use. A final opinion on the matter is left to the editor.

GG: Currently, there are many different scripts available with which we are doing the processing. These are however not well enough documented yet to be stored in an open repository. We are further working on the topic and finally want to provide some re-usable scripts. For the time being we would like to offer that the interested reader can contact us and we provide what is available.

Minor points:

- Lines 120-121: traditionally, Marshall-Palmer is used to indicate a power law relation in which the parameters are 200 and 1.6 (e.g. see Uijlenhoet, 2001, https://doi.org/10.5194/hess-5-615-2001). I suggest to use a term such as "power-law" relation or similar.

GG: adapted

- Line 125: some additional details on the merging would be helpful. What data is used? How is the merging handled in ungauged locations? The reference is there but some basic details are needed - especially since you later provide the validation, which makes me think this operation is done on this data by this study and not in the reference.

GG: adapted. See 2.2.2. last paragraph in the final version.

- Figure 2:

- what return level is shown in the figures? The caption does not say it. Is this behaviour consistent with across return levels?

GG: corrected

- Caption: These are technically DDF (not IDF) as the displayed value is rainfall depth

GG: corrected

- The two examples shown in the figure display an overestimation of the radar values. This is different from what reported by several studies and in agreement with some other (e.g. see the references in Marra et al 2019, https://doi.org/10.1016/j.jhydrol.2019.04.081). I think a comment on the potential sources of this overestimation should be provided.

GG: Our experience has shown, that looking at long term averages radar data often underestimate rainfall compared to station observations. However, looking at extremes we found often the opposite. We are not sure for the reasons. Although the data is corrected the overestimation can a result of the attenuation correction. Radar data validation is not a focus of this paper. The sections on radar data are added to support the validity of the data used and the merged data is the one which is used for the analysis. The corrected radar data results are provided as a comparison to show the improvements resulting from the merging.

- Equations 2: it seems a division by n_gauge is missing (I guess this wants to be the average?)

GG: corrected

- Equation 3: it seems a division by n_gauge is missing from inside the square root and from the denominator. If not, I believe the metric is not actually what one expects as a normalised RMSE
GG: corrected

- Figure 3: it would be interesting to check whether there is any systematic deviation of RMSE and Bias with return period. The boxplots now merge all the probabilities and do not allow for these interpretations

GG: adapted the plot so the return periods are distinguishable. Considering the pBias the error increases consistently with return period in both products, the corrected radar and the merged product. The higher the return period the less significant the increase. Also, for the merged data the increase of the error measure with return period is less significant than in corrected radar data. For nRmse on the other hand, at shorter durations the error decreases with return period.

- Lines 143-144: I suggest to explain the meaning of these metrics (e.g. RMSE is 20% of the estimated value and percent bias shows ~6% underestimation)

GG: Done

- Line 161-162: please refer to section 3.1 where the method is presented.

GG: Done

- Line 163-164: I am not sure the plotting is part of the ADDF computation procedure. Perhaps this part should not be a numerated item.

GG: corrected

- Lines 170-179: it seems the subscript a in i_a,d is lost somewhere between eq 8 and 9. Is each area treated independently and Koutsoyianis method is used only for handling multiple durations? A reader not familiar with this method would probably get lost here.

GG: corrected. Not the equation, but how the text leads up to it. Hopefully it is easier to understand now?

- Lines 186-186: this is indeed the advantage of such an approach. I guess this also comes with some limitations as annual maxima from many durations are highly correlated, so the actual information contained in the data is less than what it would be in case of independence. This is likely enhanced by the inclusion of the areal averaging. I believe a comment could be useful here.

GG: Done

- Lines 187-188: please explain why the method of the L-moments is used. Usually it is preferred in the case of limited data samples, but after the Koutsoyiannis normalization the sample becomes relatively large. Is it a matter of computation costs with respect to maximum likelihood, or is it still a matter of sample numerosity?

GG: I added:

*The method of L-moments is chosen here over the Maximum Likelihood method per suggestion of Shehu et al. (2023), due to the more stable results of the moments.*